# Patient-Derived Organoids as a Promising Tool for Multimodal Management of Sarcomas

**DOI:** 10.3390/cancers15174339

**Published:** 2023-08-30

**Authors:** Songfeng Xu, ShihJye Tan, Ling Guo

**Affiliations:** 1National Cancer Center/National Clinical Research Center for Cancer/Cancer Hospital & Shenzhen Hospital, Chinese Academy of Medical Science and Peking Union Medical College, Shenzhen 518116, China; xusongfeng@yeah.net; 2Department of Orthopedics, National Cancer Center/National Clinical Research Center for Cancer/Cancer Hospital & Shenzhen Hospital, Chinese Academy of Medical Science and Peking Union Medical College, Beijing 100021, China; 3Guangdong Provincial Key Laboratory of Cell Microenvironment and Disease Research, Department of Biology, Academy for Advanced Interdisciplinary Studies, Southern University of Science and Technology, 1088 Xueyuan Blvd, Biology Building 402, Shenzhen 518055, China

**Keywords:** sarcomas, patient-derived organoids, personalized medicine

## Abstract

**Simple Summary:**

Patient-derived sarcoma organoids (PDSOs) are a new and promising tool for treating sarcomas, a type of cancer that affects the body’s connective tissues. These organoids are created in the lab to mimic the complexity of a patient’s tumor. They help doctors understand the genetic and molecular makeup of the tumor and test different treatments to see which ones work best for each patient. PDSOs can guide personalized treatment decisions, ensuring that patients receive the most effective therapies tailored to their specific tumor characteristics. They also help researchers discover new biomarkers and develop better treatment options. By using PDSOs, doctors can improve patient outcomes and revolutionize how sarcomas are managed. Ongoing research is essential to fully unlock the potential of PDSOs and make them a standard part of sarcoma treatment.

**Abstract:**

The management of sarcomas, a diverse group of cancers arising from connective tissues, presents significant challenges due to their heterogeneity and limited treatment options. Patient-derived sarcoma organoids (PDSOs) have emerged as a promising tool in the multimodal management of sarcomas, offering unprecedented opportunities for personalized medicine and improved treatment strategies. This review aims to explore the potential of PDSOs as a promising tool for multimodal management of sarcomas. We discuss the establishment and characterization of PDSOs, which realistically recapitulate the complexity and heterogeneity of the original tumor, providing a platform for genetic and molecular fidelity, histological resemblance, and functional characterization. Additionally, we discuss the applications of PDSOs in pathological and genetic evaluation, treatment screening and development, and personalized multimodal management. One significant advancement of PDSOs lies in their ability to guide personalized treatment decisions, enabling clinicians to assess the response and efficacy of different therapies in a patient-specific manner. Through continued research and development, PDSOs hold the potential to revolutionize sarcoma management and drive advancements in personalized medicine, biomarker discovery, preclinical modeling, and therapy optimization. The integration of PDSOs into clinical practice can ultimately improve patient outcomes and significantly impact the field of sarcoma treatment.

## 1. Introduction

Sarcomas are rare yet complex neoplasms that originate from mesenchymal cells in various types of connective tissues, including bone, cartilage, muscle, fat, nerve, and blood vessels. There are more than 70 known subtypes of sarcoma, which can be divided into two major groups: soft-tissue and bone sarcomas [1,2]. Soft tissue sarcomas typically arise in the extremities or retroperitoneum, with most patients noticing a progressively painful tumor mass. The 5-year survival rate for individuals diagnosed with soft tissue sarcomas decreases to 15% in cases of distant metastasis [3,4]. Bone sarcomas commonly cause pain, swelling, and pathological bone fractures. Despite most bone sarcomas being detected at an early stage, the 5-year survival rate for patients diagnosed with distant stage around 30%, depending on the histological subtype [5,6]. Sarcomas can occur at any age, but certain subtypes are more prevalent in specific age groups. For instance, osteosarcoma and Ewing sarcoma are more commonly found in children and young adults [7,8], while leiomyosarcoma and malignant fibrous histiocytoma (currently classified as undifferentiated pleomorphic sarcoma by WHO) are more frequent in adults [9,10]. The causes of most sarcomas remain ambiguous, although several factors are associated with an increased risk of developing the disease, including exposure to ionizing radiation, alkylating agents, lymphedema, mutations, and inherited genetic conditions [11,12].

The rarity and heterogeneity of sarcoma subtypes have proven to be challenging factors in clinical trial accrual, as evidenced by the limited progress in sarcoma clinical outcomes following multimodality treatment [13,14]. Current treatments often have limited efficacy and significant side effects. Surgery remains the main treatment modality for the majority of sarcomas, complemented in selected cases with radiation and/or systemic chemotherapy [15,16,17,18]. Nevertheless, sarcomas frequently occur in anatomically complex locations, such as the extremities, retroperitoneum, or head and neck region, presenting challenges for surgical resection and optimal disease control. Furthermore, each sarcoma subtype possesses distinct features, including preferred age of onset, anatomical location, genetic profiles, growth pattern, tumor invasiveness, and treatment resistance [19,20] and, thus, impeding effectiveness of therapies. Although advancements in the development of cancer therapies such as immunotherapy [21,22], the rarity of sarcomas contributes to the scarcity of large-scale clinical trials and evidence-based guidelines, leading to a lack of standardized treatment protocols. The limited understanding of the underlying genetic and molecular drivers of sarcomas development and progression further hampers the improvement of prognostic prediction, and the development of targeted therapies. The heterogeneity of sarcomas within individual patients and the development of treatment resistance over time underscore the necessity for implementing multimodal management strategies to address the therapeutic limitations.

Advancements in cancer research have developed patient-derived organoids (PDOs) as a promising tool for preclinical modeling. PDOs are three-dimensional (3-D) cultivation systems of human tissues that are derived directly from patient samples. PDOs offer several advantages over traditional 2-D cell cultures and animal models as they retain the cellular diversity and architecture of the original tissue, allowing researchers to study the characteristics and behavior of patient-specific cells in a more physiologically relevant context [23,24]. Additionally, biopsy of the tumor mass is typically required for a definitive diagnosis, ensuring a readily available source for PDOs. With their expandability, cryopreservation capability, and genetic modifiability, organoids allow diverse applications in cancer research including human molecular and developmental biology, disease modeling, personalized medicine, and drug screening and resistance responses [5,25,26]. This technology presents a unique opportunity to bridge the gap between preclinical and clinical settings, addressing the challenges in sarcoma multimodal managements. This review aims to explore the potential of patient-derived sarcoma organoids (PDSOs) as a promising tool for multimodal managements of sarcomas. We will discuss the establishment and characterization of PDSOs, their applications in the study of tumor biology, drug screening, exploring novel therapeutic strategies, and predicting treatment response and personalized medicine. Additionally, we will highlight the factors required to be considered in establishing PDSOs and role of PDSOs in guiding multimodal treatment decisions. By leveraging the capabilities of PDSOs, we can strive towards improving the outcomes of patients with sarcomas and advancing the field of sarcoma management (Figure 1).

## 2. Establishment of PDSOs

The generation and establishment of PDSOs involve a series of key steps, including the acquisition of sarcoma biospecimens, tissue processing, culturing and expanding the samples in a specialized environment, and subsequent sample analysis.

### 2.1. Approaches for Obtaining Sarcoma Tissue Samples

The first crucial step in generating PDSOs is the procurement of tumor tissue from sarcoma patients. The collection of tissue samples must adhere to ethical guidelines and institutional protocols to ensure patient safety and informed consent. The acquisition of sarcoma tissue samples for organoid generation depends on various factors, including tumor location, size, accessibility, and patient-specific considerations. Common approaches for obtaining sarcoma tissue samples include tumor biopsy, surgical resection, fine needle aspiration (FNA), core needle biopsy, and explant culture [27,28]. Once the tissue samples are collected, it is essential to promptly transport them to the laboratory under appropriate conditions to effectively preserve the tumor samples. Typically, there is a time gap of a few hours between the samples being removed from the patient and their transportation to the laboratory for tissue processing. While the samples are usually placed in a medium supplemented with serum and antibiotics at room temperature, it is recommended to incubate the samples in medium at 4 °C to minimize tumor degradation and better preserve their integrity.

In the laboratory, the collected tumor sample undergoes a series of processing steps to isolate tumor cells or fragments. The specific dissociation techniques employed depend on the sarcoma subtype and tissue characteristics. These techniques can include mechanical dissociation, enzymatic digestion, or other methods to obtain a single-cell suspension or small tissue fragments. The isolated tumor cells or tissue fragments are then cultured in specialized media that provide optimal conditions for the growth and maintenance of PDSOs. The establishment of PDSOs involves culturing the isolated tumor cells in 3-D conditions. Subsequently, the generated PDSOs undergo comprehensive characterization to confirm their fidelity to the original tumor. This characterization involves assessing their genetic, molecular, histological, and functional properties. To validate the representativeness of the PDSOs, specific markers are examined, genomic sequencing is performed, protein expression is evaluated, and functional assays are conducted. These analyses allow the confirmation of the PDSOs as models that realistically recapitulate the original tumor. In the following sections, we will discuss the criteria used to ensure the heterogeneity and fidelity of PDSOs.

### 2.2. Recapitulating Tumor Heterogeneity and Complexity in PDSOs

The microenvironment of sarcomas comprises diverse cell types, extracellular matrix components, and signaling molecules, all of which play a critical role in sarcoma development and progression. For PDSOs to be effective tools in studying sarcomas, it is crucial that they closely resemble the microenvironment of the original tumor. This allows for the investigation of tumor-specific genetic events, identification of underlying molecular mechanisms, understanding the impact of the microenvironment on tumor behavior, and exploration of effective therapeutic strategies. There are various materials and techniques available for generating PDSOs. However, due to the complexity and uniqueness of each tumor, it is challenging to standardize the materials and techniques involved in the establishment of PDSOs. Researchers must carefully consider their specific research goals when selecting the most suitable materials and methods. In this section, we will discuss several key aspects, including cultivation techniques, media composition, and the involvement of different cell types (Table 1). By carefully controlling these key characteristics, tumor cells have the ability to self-organize into three-dimensional structures that closely resemble the architecture and cellular composition of the original tumor. This realistic representation of the tumor microenvironment in PDSOs enables researchers to conduct comprehensive studies and gain valuable insights into sarcoma biology.

Here, we discussed various techniques that allow the formation of organoids, including scaffold-free [29,30,31], scaffold-based [25,32,33,34,35], and xenograft-based [36,37] methods (Figure 2). Scaffold-free techniques involve the use of specialized culture plates or dishes that facilitate the self-assembly of cells into three-dimensional structures without the need for a physical scaffold. These methods are popular due to their simplicity and cost-effectiveness. However, they have certain limitations. These include potential inconsistencies in clinical cell and tissue architecture, challenges in controlling the spatial organization of cells, degradation of organoids over time, and limitations in high-throughput applications or clinical translation [38]. To address these limitations, scaffold-based techniques have been developed. In scaffold-based methods, tumor cells are embedded within a biocompatible scaffold that provides structural support and facilitates cell–matrix interactions. This approach allows for the self-organization and proliferation of cells, resulting in the formation of multicellular structures that closely resemble the architecture of the original tumor. It is important to consider material selection, mechanical characteristics (such as stiffness), and scaffold components in accordance with the research goals, as they can influence cell behavior and drug response. For instance, although Matrigel has been widely used in various cancers, studies have shown that certain sarcoma subtypes fail to form organoids in Matrigel [39,40,41] (Figure 3). This could be attributed to the inappropriate material provided, as illustrated in Figure 3, where different degrees of organoid formation are observed when cells are cultured in varying material. Currently, these material properties receive limited attention in generating PDSOs, which can lead to organoids with stiffness profiles differing from their in vivo counterparts, ultimately affecting clinical outcomes.

Xenograft-based methods involve the use of immunocompromised animals to generate PDSOs. This approach allows for the engraftment, growth, and subsequent propagation of tumor tissue from one animal to another [42]. Tumors propagated through xenograft-based methods are believed to retain tumor heterogeneity, histological characteristics, genomic profiles, and response patterns to therapies [43,44,45]. However, it is important to note that a small fraction of reports have indicated that not all xenograft-based models are genetically stable [46,47]. Additionally, there are drawbacks to this method, including the requirement for animals as hosts and the variable engraftment time, which can take several months [48]. The engraftment rate can also vary, influenced by factors such as mouse strain, tumor type, degree of tumor malignancy, and others. Engraftment rates have been reported to range from 20% to 80% [36,37,49]. For instance, Xu et al. studied xenograft PDO using thirty-six bone sarcoma and six tissue sarcoma (details listed in Table 1), and they showed an overall higher engraftment rate of 73.8% compared to most other studies, but they used a triple immunodeficient NCG mice [37]. It should be noted that the microenvironment of immunodeficient mice may not fully represent the tumor microenvironment in patients, particularly concerning immune responses and immunotherapy. Despite the limitations associated with every system, xenograft-based methods have their own advantages and can be used effectively to achieve different research outcomes, advancing our understanding of various cancer types in large cohorts, including adults, as well as pediatric and adolescent populations [43,44,45]. However, it is crucial to continue research and development in this field to optimize these methods. For example, efforts should focus on refining scaffold-free approaches to enhance reproducibility and control over organoid architecture, as well as investigating the mechanical characteristics of scaffold-based approaches to better mimic in vivo conditions. By addressing these challenges, PDSOs can serve as more accurate and reliable models for studying the disease and developing effective therapeutic strategies.

**Table 1 cancers-15-04339-t001:** Summary of methods for establishing PDSOs and their fidelity.

Method	Organoid Establishment Methods	Sarcoma Type	Coculture Condition	Culture Media	Organoid Formation Rate and Day	Molecular results—Gene Profiling, etc.	Histological and Functional Results—Morphology, Cell Growth, etc.	Treatments	Author/Year
Scaffold-free	Surgical resection, minced, digested with 0.25 U/mL liberase. Directly seeded on ultralow attachment microplate.	2 × MFS 1 × MPNST 1 × UPS 1 × ESTL	N/A	GM: advance DMEM supplemented with— 10% heat-inactivated horse serum, primocin, and 1 glutamax for MFS2. 1 glutamax, B27 supplement, primocin, 1.25 mM N-acetylcystein, 50 ng/mL rhEFG, 20 ng/mL rhFGF 10, 1 ng/mL rhbFGF, 500 nM A-83-01, 10 mM SB202190, 10 mM nicotinamide, 1 mM prosta-glandin E2, 25 nM hydrocortisone, 0.5 mg/mL epinephrine, and 50 mL R-spondin for the remaining sarcoma type. 1:1 mixture of HS:CHK at P5. SM: advance DMEM without additional growth factors	50% success rate (able to expand more than five times) 1 day spheroid formation	The PDSOs accurately reflected the mutational and copy number profiles of the original tumor tissue, with only minor changes observed in the cell models, such as in chromosome 9 from MPNST. Sarcomas commonly exhibit alterations in tumor suppressor genes like TP53 and CDKN2A, which disrupt the regulation of the cell cycle. Recurring mutations in genes such as ATM, STAG2, MSH2, MSH6, and PTEN indicate DNA repair defects that align with the genomic instability observed in sarcomas.	Morphology of spheroid mimic original tumor as determined by H&E staining. Cell doubling time ranging from 2.6 to 8 days. Cells from the same subtype of sarcoma can have different rates of cell proliferation. Cell invasion correlated to cell proliferation (cells were seeded on a special matrix (Matrigel) and a chemoattractant gradient was created to simulate cell invasion).	Tested 12 drugs. Challenged with drugs for 5 days. Gemcitabine and pazopanib showed partial responses, and carfilzomib was one of the most effective drugs. The response of the cells to docetaxel, irinotecan, cisplatin, and PU-H71 varied depending on the specific sarcoma subtype. The cells did not respond to temozolomide or ifosfamide. A combination of 1 mM doxorubicin and a low dose of carfilzomib showed a synergistic effect. The combination of carfilzomib and the HSP90 inhibitor PU-H71 had a synergistic effect on certain patients in a dose-dependent manner. There were no synergistic effects observed with the combination of carfilzomib and venetoclax.	Chen et al. 2023 [29]
	Amputation or diagnostic biopsy, minced, digested with 1 mg/mL liberase. Directly seeded on ultra-low attachment microplate.	2 × EMC	N/A	1:1 advanced DMEM supplemented with 10% heat-inactivated horse serum, Glutamax, 100 μg/mL primocin mixed with CHK Media containing, B27 supplement, 1.25 mM N-acetylcysteine, 50 ng/mL human recombinant EGF, 20 ng/mL human recombinant FGF-10, 1 ng/mL recombinant human FGF-basic, 500 nM A-83-01, 10 μM SB202190, 10 mM Nicotinamide, 1μM PGE2, 25 nM Hydrocortisone, and 0.5 μg/mL epinephrine and R-Spondin.	N/A	Both the tumor and PDSOs had a low tumor mutational burden and a stable microsatellite status. The copy number profiles exhibited gains mainly in chromosome 1 and 8 for patient with and without lung metastasis, respectively. The STR (short tandem repeat) patterns of the cells did not match any other cell lines available in public cell banks, as checked using the cell line database.	Phenotypic analysis by H&E staining showed morphology of PDSOs recapitulate the native tumor. 5.09 days and 6.05 days of cell doubling duration for patient with or without lung metastasis.	Tested 40 drugs. Started challenging the cells with drugs 24 h after placing them and continued for 6 days. Carfilzomib was highly effective, followed by doxorubicin, which showed good effectiveness. Other chemotherapeutics showed moderate, low, or no effectiveness. PU-H71 (HSP90) and HDM201 (MDM2/MDM4) performed best from the compounds tested and showed good sensitivity while the cells had a moderate sensitivity to venetoclax. The combination of carfilzomib with venetoclax and doxorubicin showed dose-dependent synergistic effects in patient-specific manner.	Bangerter et al. 2023 [30]
Scaffold-based	Resection, minced and digested with collagenase HA and BP protease. Hyaluronic acid/gelati	AS LMS GST LS MFS DFSP PAS	1:3 tumor cells to immune cells (iPDSO).	DMEM with 10% FBS with 1% penicillin–streptomycin, and 1% L-glutamine.	100% success rate 7 days of organoid formation	MMP14, MMP 2, and MMP9 varied significantly between immortalized versus non-immortalized cells. A striking difference in LAMB3, where the immortalized cells show decreased expression compared with the elevated profile of non-immortalized cells.	Hematoxylin and eosin (H&E) stain, both the PDSOs and matched patients’ tumor showed atypical spindle cell proliferation in LMS, LS, and DFSP patients. Both PDSOs and patients’ tumor displayed similar expression of markers, including Ki67, CD34, PARP1, VIM, and MMP9.	Tested 10 drugs. Challenged on day 7 following organoid biofabrication for 72 h. About 33% of the sarcoma PDSOs did not respond to chemotherapy, while a smaller group showed a response to immunotherapy, which aligns with what is seen in clinical practice. A minority of the specimens showed a drug-, tumor-, and patient-specific response to immunotherapy. None of the iPDSOs were sensitive to ipilimumab. The original DFSP1 tumor PDSO cells responded to imatinib, doxorubicin, and regorafenib, while the immortalized tumor cells became resistant to regorafenib at passage 12. The non-immortalized cells remained sensitive to imatinib but became resistant to doxorubicin at passage 8 and regorafenib at passage 12.	Forsythe et al. 2022 [25]
	Surgical resection, minced, and digested with 50 mg/mL of liberase. Collagen type 1 gels according to the air–liquid interface organoid culture method Matrigel +NOD-scid IL2Rgnull (NSG) mouse	2 × ES	N/A	Advanced DMEM/F12 supplemented with 10 mM HEPES, GlutaMAX, and Penicillin–Streptomycin–Glutamine.	N/A	The gene profile of the original tumor showed moderate similarities to the profiles of PDSO. This could be due to the presence of different types of normal cells in the original tumor or changes in the genetic characteristics of the organoid cells. The gene expressions were grouped into four clusters, and cluster C was found in both the original tumor and ODXs. In this cluster, 225 genes and 15 biological pathways were identified, including those involved in cancer-related proteoglycan, cell cycle regulation, and the P53 signaling pathway.	PDSOs displayed a histological appearance that closely resembled the original tumor and remained intact throughout the culture and xenografting process. PDSOs showed positive immunoreactivity for pan-keratin, vimentin, CD34, and PCNA, indicating similarities to the original tumor. Cell doubling 2.3 to 3.3-fold in 72 h	The organoids showed sensitivity to doxorubicin within the range of 1.5–15 mM for both patients. The response to paclitaxel varied depending on the patient, with a sensitivity range of 0.04–4 mM.	Wakamatsu et al. 2022 [32]
	Needle biopsies or resection specimens, minced. The pieces were placed on a strainer and scraped to create a single-cell suspension, which was collected in a tube for 2-D cell culture. The remaining tissue fragments on the strainer were collected in another tube for 3-D cell culture. Basement membrane extract	19 × RMS	N/A	BM: Advanced DMEM/F12 supplemented with Glutamax, Penicillin/Streptomycin, and B27 (without vitamin A) Base medium supplemented with N2, N-acetylcysteine (500 mM), MEM nonessential amino acids, Sodium pyruvate (100 mM), Heparin (5000 U/mL), hEGF (2 lg/mL), hFGF-basic (40 lg/mL), hIGF1 (100 lg/mL), RKi (Y-27632, 100 mM), and A83-01 (5 mM) Advanced DMEM/F12 supplemented with Glutamax, Penicillin/Streptomycin (10,000 U/mL), and HEPES (1 M)	41% of success rate 4–8 weeks for tumoroid formation	Various methods, including RT-qPCR, whole-genome sequencing for copy number profiles and mutational signatures, assessment of individual genetic variations, and comparative transcriptomic analyses, indicate that the RMS tumoroid models closely resemble the original RMS tumors they originated from. Single-cell RNA sequencing (scRNA-seq) reveals that the RMS tumoroid models maintain diversity in the expression of specific genes (such as MYOG, MYOD1, and DES). Notably, this diversity is not influenced by differences in cell cycle activity, as gene expression patterns do not align with the cell cycle marker MKI67.	Molecular and histological analysis confirms that the models closely resemble the original tumors and remain genetically stable even after being cultured for up to 6 months. Specific histopathology markers, such as DES, MYOG, MYOD1, and fusion transcripts in fusion-positive RMS (17 out of 19 cases), have been successfully identified and established in the models.	The average time for drug screening in these models is around 81 days. The drug screening results accurately reflect the known sensitivities of the tumors, and the models can be modified using CRISPR/Cas9 technology to study the effects of specific genetic alterations, such as TP53 knockout. The proteasome inhibitor bortezomib also shows high efficacy in all models, consistent with previous studies that demonstrated its effectiveness in RMS tumors both in the laboratory and in animal models. This suggests that the RMS tumoroid models accurately reflect the drug sensitivities observed in RMS tumors.	Meister et al. 2022 [33]
XenograftODX	Biopsy, surgical resection, blood or bone marrow sampling 3–7 weeks immunocompromised nude mice	50 × ES 25 × RMS (2 × FOXO1 fusion-positive aRMS, 9 × eRMS and one each epithelioid, sclerosing, spindle cell, and NOS)	N/A	N/A	0–82% of success rate	All aggressive RMS xenografted PDSOs retained the PAX3::FOXO1 fusion gene, as well as other genetic alterations associated with aggressive features like TP53 mutations, FGFR4 mutations, PIK3CA or NRAS mutations, and alterations in cell cycle-related genes. Fifteen Ewing sarcoma xenografted PDSOs were established, with 13 of them showing the EWSR1::FLI1 translocation, and one each showing FUS::ERG and EWSR1::FEV translocations. The xenografted PDSOs also exhibited other genetic alterations such as TP53 mutations, STAG2 mutations or deletions, and CDKN2A deletions.	All exhibit the typical small round cell morphology. P0 to P1 took 12–285 days, and P1 to P2 took 5–104 days.	N/A	Costa et al. 2023 [36]
	Surgical resection with wide margins and arthroplasty Air–liquid interface organoid culture was xenografted into 6-week-old NOD-scid IL2R mice after serial passage	1 × MGT	N/A	GM	30–60 days for average outgrowth time	Genetic changes found in MGT as compared to giant cell tumor of the bone, where 15 of 26 mutations were found in the selected genes, including oncogenic mutations of TP53 and TSC2. Gene profile of MGT showed moderate correlation with giant cell tumor of the bone, while gene profiles of xenograft-derived organoid almost identical to original tumor.	Air–liquid interface-cultured organoid showed dendritic morphology. Xenograft-derived organoid histological appearance resembling that of the original tumor. Air–liquid interface-cultured organoid increased 2.5 fold in 72 h.	N/A	Suzuki et al. 2023 [50]
	Surgical resection or biopsy, primary lesion or metastasis. 6–8 weeks immunocompromised nude mice	30 × OS 3 × CS 2 × CD 1 × GS 6 × SS 1 × RMS 1 × MS 1 × HS	N/A	N/A	73.8% of success rate 57.5 days of average outgrowth (Independent of age and sex)	The most commonly altered genes among the top five mutations were P53 and KMT2C, followed by APOB, BRCA2, and KRAS. Patients with metastasized tumors had a higher proportion of alterations in APOB, BRCA2, FAT1, LATS1, and SPTA1 genes. On the other hand, KMT2C, NIPBL, TP53, AXIN2, and SETBP1 were more frequently mutated in tumors without metastasis. Patients with unfavorable prognoses (death or recurrent tumor) had more frequent mutations in KMT2C, ARHGAP35, JAK3, and KRAS genes, while RB1 and LATS1 mutations were more common in patients who remained event-free (without recurrence) within one year.	All tumour pairs displayed a high consistency histopathologically.	The combination of docetaxel and gemcitabine showed a response in 9 out of 12 cases. Paclitaxel was the next most effective drug, with 7 out of 12 cases responding to it. Apatinib, an anti-angiogenesis agent, had the lowest response rate, with only one patient showing a response. The predictions made by patient-derived tumor xenografts (PDTX) matched the actual responses of the patients in all cases (100% concordance). The study mentioned three cases where patient-derived tumor organoids (PDSOs) were used to guide clinical treatments.	Xu et al. 2022 [37]

Abbreviations: base medium (BM), growth media (GM), screening media (SM), myxofibrosarcoma (MFS), malignant peripheral nerve sheath tumor (MPNST), undifferentiated pleomorphic sarcoma (UPS), extrauterine soft tissue leiomyosarcoma (ESTL), extraskeletal myxoid chondrosarcoma (EMC), angiosarcoma (AS), gastrointestinal stromal tumor (GST), liposarcoma (LS), dermatofibrosarcoma protuberans (DFSP), pleiomorphic abdominal sarcoma (PAS), epithelioid sarcoma (ES), rhabdomyosarcoma (RMS), malignant giant-cell tumor (MGT), chondrosarcoma (CS), chordoma (CD), giant osteosarcoma (GOS), synovial sarcoma (SS), malignant schwannoma (MS), hemangiopericytoma (HP), metalloproteinases (MMP9), Poly [ADP-ribose] polymerase 1 (PARP1), vimentin (VIM), hyaluronic acid (HA), immunohistochemistry (IHC), and not applicable (N/A).

In addition to tumor architecture, the presence of heterogeneous cell populations within the tumor can lead to variations in sensitivity or resistance to treatments, posing a challenge for effectively targeting all cell subtypes (Table 2). Sarcoma subtypes and individual tumors can exhibit variations in the proportions, spatial arrangement, morphology, gene expression, and functional characteristics of these cell populations. While conventional cell lines are easy to use, cost-effective, and allow large-scale screening and genetic modification, they often fail to recapitulate many basic features of the genetic and molecular background of the tumor they were derived from, limiting their predictive value [51]. PDSOs overcome the limitations of conventional cell lines and strive to capture the heterogeneity of the original tumor by maintaining diverse cell populations, including cancer stem cells, differentiated cells, and stromal cells. This enables the study of tumor subpopulations and their interactions, providing insights into intra-tumoral heterogeneity and personalized treatment strategies. However, it is important to note that PDSOs primarily preserve cells within the tumor and may overlook the contribution of the tumor microenvironment, including immune cells and cells outside the tumor. Considering these factors becomes crucial when establishing PDSOs depending on the research goals. For instance, co-culturing PDSOs with immune cells can lead to a better understanding of prognosis and the development of immunotherapy strategies. Moreover, various tissue engineering methods such as micropatterning and organ-on-chip can be utilized in establishing a complete PDSO system.

Various methods have been employed to expedite the formation of organoids and establish long-term, renewable PDSOs. However, it is important to acknowledge that these techniques can introduce alterations to the PDSOs. For example, studies have demonstrated that cell expansion and the use of immortalized cell lines can lead to genetic and functional changes, which can significantly affect the reliability and accuracy of in vitro treatment responses [52,53]. Moreover, specialized media used for culturing PDSOs are often supplemented with growth factors, cytokines, and other components that are non-clinical in nature (Table 1) [29,30,33]. These factors have the potential to influence cellular responses such as proliferation, differentiation, and drug resistance, thereby impacting the accuracy of patient outcomes [29]. In the past, the effects of these factors, particularly in functional assays like drug screening, were often overlooked by researchers. To achieve a clinically relevant setting, it is crucial to develop media formulations that closely mimic the physiological microenvironment of sarcomas [29,33]. Additionally, culture conditions such as temperature, humidity, and gas composition must be carefully controlled to replicate the physiological conditions found in the tumor microenvironment. Standardizing the establishment of PDSOs remains a challenge; however, the successful generation of PDSOs holds tremendous promise for advancing sarcoma research and improving clinical practice.

### 2.3. Molecular, Histological, and Functional Fidelity of PDSOs

A well-established PDSO should accurately recapitulate the heterogeneity and complexity of the original sarcoma tumor. This includes assessing its histological features, genetic and molecular profiles, and functional properties [33,54,55].

#### 2.3.1. Molecular Fidelity

Molecular fidelity encompasses the preservation of genetic and post-translational profiles, modifications, and events that play crucial roles in gene expression, mutations, epigenetic modifications, cellular signaling, protein–protein interactions, protein degradation, and other biological processes [55,56]. By retaining the molecular features, organoids provide a representative platform to study the molecular drivers of sarcoma development, progression, and treatment response. Molecular analysis techniques such as genomic sequencing, transcriptomic profiling, and immunohistochemistry are employed to validate the molecular fidelity of the PDSOs, ensuring that they retain the genetic and molecular characteristics of the original tumor [25,29,36].

#### 2.3.2. Histological Fidelity

Histological fidelity ensures that organoids resemble the intra-tumoral heterogeneity of the original tumor, facilitating accurate modeling and analysis [56,57]. Histological fidelity involves the preservation of histological features, cellular composition, tissue architecture, and differentiation patterns of the original tumor [58,59]. The organoids should maintain the presence of diverse cell types and microenvironmental settings with similar morphological and structural properties as observed in the original tumor. This allows the study of cell subpopulations, morphology, cellular interactions (e.g., cell–cell and cell–matrix), and the contribution of cellular heterogeneity to sarcoma biology [60]. Histological evaluation involves staining the PDSOs with specific dyes and performing microscopic examination to determine if they resemble the original tumor’s histological features.

#### 2.3.3. Functional Fidelity

Functional fidelity refers to the ability of PDSOs to recapitulate key functional properties of the original tumor. This includes maintaining the synthesis of small molecules (e.g., inflammatory cytokines and MMPs), proliferative capacity, invasive potential, and response to therapeutic agents [61]. Functional fidelity allows for the evaluation of drug sensitivity, identification of effective treatment strategies, and prediction of patient response [5]. By maintaining functional properties, organoids provide a valuable platform for studying the behavior and characteristics of sarcomas in a controlled laboratory setting.

Overall, the generation and establishment of PDSOs involve the collection of tumor tissue, isolation of tumor cells, carefully controlled laboratory conditions, and optimization of culture protocols to promote the growth and maintenance of organoid structures that closely resemble the original tumor. PDSOs serve as powerful tools to bridge the gap between laboratory research and clinical practice, allowing researchers to study the underlying biology, test novel therapies, and develop personalized treatment approaches for sarcoma patients. A well-established PDSO should accurately recapitulate heterogeneity and complexity of the original sarcoma tumor. This includes assessing their histological features, genetic and molecular profiles, and functional properties.

## 3. PDSOs in Multimodal Management of Sarcomas

### 3.1. Pathological and Molecular Genetic Evaluation

PDSOs offer valuable insights into the complex mechanisms underlying sarcoma heterogeneity, development, progression, and metastasis. These organoids realistically replicate the intricate nature of the original tumor, encompassing its genetic and cellular characteristics. Through the analysis of whole-genome sequencing, RNA sequencing, and single-cell sequencing in PDSOs, researchers can gain a deeper understanding of disease mechanisms, tumor advancement, and metastatic behavior (Figure 4). For example, Maurer et al. developed Ewing sarcoma PDOs from a metastatic pulmonary lesion in a patient with an inherited BRCA1-associated RING domain 1 (BARD1) mutation [62]. Their study revealed that the loss of BARD1 increases Ewing sarcoma sensitivity to DNA damage, with Guanylate-binding protein 1 playing a role in the DNA damage response within Ewing sarcoma organoids. This discovery sheds light on important mechanisms involved in Ewing sarcoma [62]. Studying these underlying mechanisms not only enhances our understanding of tumors but also facilitates the identification of specific biomarkers and tailored treatments. In another study, Boulay et al. conducted genome-wide epigenomic profiling in synovial PDSOs, uncovering distinctive patterns of BRG1/BRM-associated factor (BAF) complex distribution [31]. Broad BAF complex domains were found to correlate with active chromatin states and the expression of a tumor-specific gene signature [31]. The BRG1/BAF complex and its interaction with the SS18-SSX fusion protein emerged as significant molecular features in synovial sarcoma [31]. Additionally, the study revealed that synovial sarcoma cells exhibited higher sensitivity to the ubiquitin-specific protease 7 inhibitor FT827 compared to Ewing sarcoma cells [31], suggesting potential treatment resistance in different subtypes.

**Figure 4 cancers-15-04339-f004:**
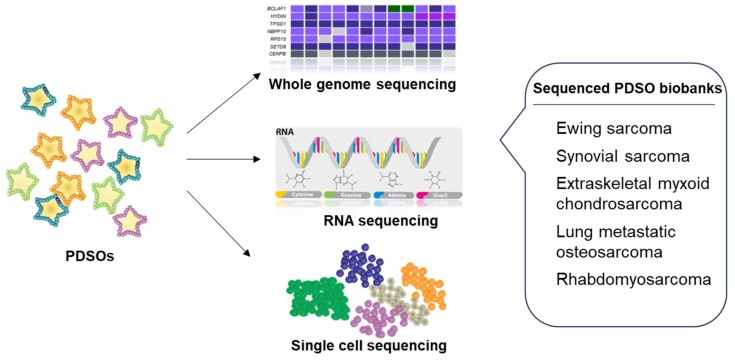
Molecular characterization by sequencing. PDSOs were characterized by whole-genome sequencing, RNA sequencing, or single-cell sequencing [30,31,32,33,36,37,62,63].

By studying the genetic and growth patterns of PDSOs, researchers can gain a deeper understanding of tumor progression and metastatic potential. For instance, Bangerter et al. created extraskeletal myxoid chondrosarcoma PDOs from patients with and without lung metastasis [30]. The study revealed differences in cell doubling times and indicated distinct genetic rearrangements associated with lung metastasis [30]. These findings suggest that genetic evaluation of PDSOs could be utilized to predict the aggressiveness of the tumor. PDSOs also offer opportunities to study secondary sarcomas. He et al. generated a PDOs platform for lung metastatic osteosarcoma, preserving the cellular morphology and expression of osteosarcoma markers [63]. Notably, the primary lung metastatic osteosarcoma organoids retained the T-cell distribution of the parental tumors, and anti-programmed cell death protein 1 (PD1) treatment activated CD8+ T-cells in the organoid cultures [63]. Furthermore, Meister et al. employed CRISPR/Cas9 to genetically edit rhabdomyosarcoma cells, demonstrating the increased sensitivity of p53-deficient rhabdomyosarcoma cells to the checkpoint kinase inhibitor prexasertib [33]. The findings highlight the significant contribution of PDSOs to the advancement of molecular research, offering valuable insights into essential pathways and molecular targets that can be specifically targeted for therapeutic interventions. By elucidating the underlying mechanisms, PDSOs not only facilitate the development of targeted therapies but also enable the identification of biomarkers for early disease detection and facilitate disease monitoring throughout the multimodal treatment journey.

### 3.2. Drug Screening and and Development of Effective Treatment

Drug screening plays a crucial role in the development of effective therapies for sarcomas, a diverse group of rare cancers arising from mesenchymal tissues. PDSOs have emerged as valuable tools for drug screening. This approach offers several advantages over traditional 2-D cell cultures as it recapitulates the complexity and heterogeneity of the original tumor, and it is higher in throughput and lower in cost compared to animal models. PDSOs can be established as fast as in a single day and allow automated administration for various drug tests in a single plate (Figure 5).

For instance, Bangerter et al. created myxoid chondrosarcoma PDOs and administered 40 drugs of various concentrations individually 24 h after plating [30]. They found that myxoid chondrosarcoma PDOs showed high sensitivity to carfilzomib and good-to-moderate sensitivity to doxorubicin, while there was no response to venetoclax as a monotherapy in the validation [30]. PDSOs for drug testing are not limited to one subtype of sarcoma. In another study, Chen et al. established and identified myxofibrosarcoma, malignant peripheral nerve sheath tumor, undifferentiated pleomorphic sarcoma, and extrauterine soft tissue leiomyosarcoma that sensitively responded to carfilzomib [29]. The creation of thoroughly characterized PDSOs provides an opportunity to evaluate novel drug compounds and explore potential synergistic combinations. For instance, Chen et al. found a synergistic effect of doxorubicin in combination with carfilzomib [29]. It should be noted that the combination of carfilzomib and doxorubicin has been widely explored, and drug synergy has been shown in 40 patients with multiple myeloma in clinical trials (NCT 01246063) [64,65]. Furthermore, Gaebler et al. successfully created long-term non-rhabdomyosarcoma organoids derived from patients with myxoid liposarcoma, undifferentiated pleomorphic sarcoma, or biphasic synovial sarcoma [66]. They used a multiplexed protein-profiling assay for drug screening, providing insights into (phospho)-proteomics.

The PDSO platform not only allows high throughput drug screening, but it can also be utilized for determining the effectiveness of various drugs in inhibiting various sarcoma subtypes. For example, Maloney et al. developed skin fibrosarcoma PDOs and subjected them to the tyrosine kinase inhibitor imatinib or the antibiotic anthracycline chemotherapy agent doxorubicin [35]. They observed a significant decrease in adenosine 5′-triphosphate (ATP) activity only after the organoid cultures were exposed to a high concentration of imatinib, while low concentrations of doxorubicin resulted in a significant reduction in ATP activity [35]. PDSOs not only suggest the effectiveness of different drugs in inhibiting various sarcoma subtypes but also allow the assessment of treatment toxicities and resistance mechanisms. This model enables researchers to identify compounds that preferentially target cancer cells rather than healthy ones, leading to the selection of less toxic substances and reducing the risk of side effects. Johansson et al. created osteosarcoma PDOs and demonstrated that the organoids preserved similar resistance profiles as the cryopreserved cancer cells from the original tumor [67], suggesting that PDOs can be used to study and identify drug resistance. Other treatments can also be employed in PDSOs. For example, Veys et al. created SW1353-cell-derived chondrosarcoma organoids to test the anti-tumor activity of microRNA-342-5p and microRNA-491-5p and concluded that microRNA-342-5p significantly promotes apoptosis, especially in hypoxia [68]. This suggests that, in addition to the use of drugs, microRNAs could be potential treatments for chondrosarcoma.

**Figure 5 cancers-15-04339-f005:**
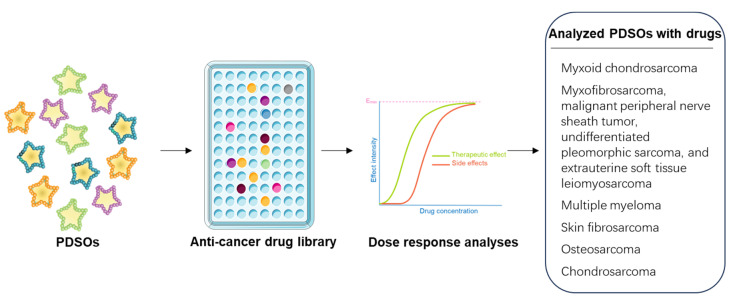
High-throughput drug screening. The application of high-throughput drug screening assays on generated PDSOs, utilizing anti-cancer drug libraries of varying capacities, and facilitates the analysis of dose response in the PDSOs [29,30,35,37,64,65,67,68].

### 3.3. Personalized Multimodal Management

Sarcoma organoids offer the potential to develop personalized treatment strategies for individual patients. Studies have demonstrated high genetic heterogeneity despite the same histologic subtype. For instance, myxofibrosarcoma obtained from different patients showed diverse molecular and functional characteristics such as cell doubling rate, invasion capacity, and drug responses [29]. The author proposed that the variation in growth rate and invasiveness may be partly explained by the presence of mutations in cell cycle checkpoint genes, such as CDKN2A, which is found to be mutated in approximately 16% of myxofibrosarcoma cases [29]. The loss of CDKN2A is associated with high-grade tumors that recur and progress [29]. A faster growth rate and invasiveness could result from the loss of this tumor suppressor CDKN2A in a patient-derived myxofibrosarcoma [29]. The study not only indicated the importance of personalized medicine but also emphasized the need for researchers to consider this factor in the development of new treatments. By testing the response of PDSOs to different drugs, clinicians can make informed decisions about the most effective treatment options, tailoring therapies to each patient’s specific tumor characteristics. The biofabrication of organoids for drug testing can typically be accomplished in around 7 days, enabling researchers to gather data on treatment effectiveness as early as 10 days after obtaining a tumor biopsy or following surgical resection [52,53]. This rapid timeline enables the selection of a rational treatment regimen for patients without significant delays in treatment initiation.

By realistically replicating the characteristics of the original tumor, precision medicine platforms offer a valuable opportunity to understand individual drug responses and customize treatments based on the unique molecular and phenotypic profiles of patients. These platforms enable the seamless translation of drug screening outcomes back to the patient, indicating the potential for incorporating the use of patient-derived organoids (PDOs) as a standard practice in personalized clinical care (Figure 6). Xu et al. reported a few cases utilizing PDSOs for personalized multimodal management to improve clinical outcomes [37]. For example, in one case, a 16-year-old male diagnosed with synovial sarcoma initially received a post-surgical systemic adjuvant therapy with doxorubicin [37]. The PDSOs model allowed clinical improvement by suggesting liposomal-doxorubicin as a better agent that have same efficacious but lower toxicity as compared to doxorubicin [37]. The researchers administered liposomal-doxorubicin sequentially to the patient, resulting in over 20 months of event-free survival [37]. PDSOs allowed clinicians to improve therapy by replacing the initial regimen with an agent of the same efficacy but lower toxicity. Another case involved a 37-year-old male primarily diagnosed with highly differentiated non-classical osteosarcoma at the 12th vertebra [37]. The patient developed drug resistance after undergoing four rounds of surgery and eight rounds of alternative chemotherapies (doxorubicin + cisplatin and methotrexate + ifosfamide) [37]. PDSOs were established during the fourth round of surgery, and they predicted a low efficacy of apatinib and methotrexate, which aligned with the patient’s previous drug resistance profile [37]. PDSOs also predicted that etoposide + ifosfamide and gemcitabine + docetaxel would have high efficacy [37]. Consequently, the patient switched to etoposide + ifosfamide for two rounds of therapies and gemcitabine + docetaxel for an additional six rounds [37]. During those therapies, the patient did not experience further disease progression and achieved over 1 year of event-free survival [37]. These typical cases illustrate the value and practicality of PDSOs in clinical settings and in the multimodal treatment of sarcomas.

In the emerging era of personalized medicine, immunotherapy has become a therapeutic option in many cancers. Its application in sarcomas is being explored but is currently recommended only for tumors with increased PD1 and PDL1 expression, deficiency in mismatch repair proteins, or increased tumor mutational burden (TMB) [69]. Although studies have demonstrated responses with combination immunotherapy in unresectable or metastatic angiosarcoma and leiomyosarcoma subtypes, extensive clinical data on immunotherapy efficacy in these specific sarcoma subtypes do not exist and are unlikely to be generated from cohort analysis [70]. This may be due to how realistically the PDSOs have recapitulated the original sarcoma. As mentioned in the section on the establishment of PDSOs in this review paper, tumor microenvironmental factors, such as immune cells, should be taken into consideration. Forsythe et al. reported on a hydrogel-based PDSOs system for evaluating chemotherapy efficacy across various sarcoma subtypes while integrating immune cells [25]. They have developed an advanced immune-enhanced patient-derived sarcoma organoid (iPDSO) system that incorporates patient-matched immune cells, serving as a tool for testing immune checkpoint blockade and guiding the development of personalized immunotherapy [25]. The availability of a platform to assess therapies targeting various components of the immune-tumor microenvironment, including T cells, natural killer (NK) cells, antigen-presenting cells, B cells, and regulatory T cells, has the potential to revolutionize the treatment of tumors that were previously not targeted by immunotherapy due to a lack of actionable targets [71,72]. While the iPDSOs platform holds potential for preclinical validation of novel therapeutic agents or combination therapies, the extent to which the PDSOs in this study align with patient outcomes remains uncertain due to the unfeasibility of directly correlating clinical data with patient results. This is currently not within sarcoma treatment algorithms due to a paucity of clinical efficacy data. Radiation and/or other treatments can also be incorporated into the study of PDSOs to determine the ideal treatment for targeted patients.

### 3.4. Challenges and Limitations of PDSOs in Multimodality Managements

While PDOs hold great promise for the management of sarcomas, there are several limitations and challenges that need to be addressed. Addressing these challenges and limitations will be key to harnessing the full potential of PDOs in improving sarcoma management and personalized treatment strategies. These challenges include tumor heterogeneity and sampling bias, tissue availability and viability, standardization and scalability, genetic stability and evolution, limited functional complexity, and clinical translation and validation. Sarcomas are highly heterogeneous tumors with diverse subtypes and genetic profiles, which may not be fully represented by PDOs due to sampling bias. To mitigate this limitation, careful selection of representative tumor regions and multiple PDSOs from different areas can be utilized. Generating PDOs that accurately capture the heterogeneity of sarcomas remains a challenge, and establishing representative organoids that encompass the full spectrum of tumor cell populations is crucial for reliable drug screening and personalized medicine approaches. Obtaining sufficient and high-quality sarcoma tissue samples for PDO generation can also be challenging. The success of PDO establishment relies on the availability of fresh tumor samples, which may be limited in some cases. Additionally, the viability of sarcoma tissue during the isolation and culture process can vary, impacting the success rate of PDO generation.

Sarcoma PDOs require specific materials and techniques to mimic the tumor microenvironment and maintain their viability and functionality. Standardization of protocols is crucial to ensure reproducibility and scalability, facilitating widespread adoption of PDOs in research and clinical settings. However, establishing standardized protocols that can be easily reproduced across different laboratories is still a challenge. PDSOs are susceptible to genetic changes and alterations during long-term culture, which can lead to discrepancies between PDSOs and the original tumor. This can affect their relevance for drug screening and therapeutic development. Monitoring and characterizing PDSOs continuously or perfecting the tumor microenvironment are necessary to ensure their genetic stability and fidelity. While PDOs recapitulate many aspects of the original tumor, they still lack the full complexity of the tumor microenvironment, including interactions with immune cells, stromal cells, and blood vessels. This limited functional complexity may impact the response to certain therapies and the ability to accurately model tumor–immune interactions. Further investigations are needed to address these weaknesses and enhance the predictive power of PDSOs for patient therapies. Moreover, the turnaround times for obtaining specific molecular alteration information through next-generation sequencing (NGS) assays are currently lengthy, limiting their usefulness in informing treatment decisions. Efforts are needed to expedite NGS results and extend the viability of reliable PDOs in culture.

The successful translation of findings from PDSO studies into clinical practice requires rigorous validation and clinical trials. Demonstrating the clinical utility, predictive power, and reproducibility of PDSO-based approaches can be challenging. Bridging the gap between preclinical research using PDSOs and their integration into clinical decision-making is crucial. While there is a rapidly increasing volume of data where organoids employed either in the setting of prospective randomized trials or retrospective analysis demonstrated 100% negative predictive value in a variety of different primaries, positive predictive value, specifically the ability to predict treatment efficacy, remains a challenge for organoid models. Additionally, the differences between cells cultured in PDSOs and the native tumor may weaken their representativeness and potential for predicting treatment efficacy.

## 4. Conclusions and Future Perspective

The advancements of PDSOs in multimodal management have revolutionized the field of sarcoma treatment. These organoids provide a powerful platform for diverse applications in drug screening, personalized medicine, treatment response prediction, incorporating immunotherapies, mechanistic studies, biomarker discovery, combination therapies, preclinical modeling, biomaterial testing, and therapy optimization. The use of PDOs holds great potential for improving patient outcomes and driving advancements in sarcoma management. However, the field of PDOs in the management of sarcomas is still relatively young and evolving, necessitating continued research and development. Ongoing research is crucial for enhancing the fidelity and applicability of sarcoma PDOs. Collaborative efforts among researchers, clinicians, and industry partners are essential for addressing the limitations and challenges associated with PDSOs. Robust databases and repositories of sarcoma PDOs, along with associated clinical and molecular data, can accelerate research progress. Sharing resources and knowledge can lead to a collective understanding of sarcoma biology and the development of standardized protocols and guidelines for PDO-based studies. Advancements in culture techniques, optimization of conditions, and standardized protocols are needed to improve the reproducibility and scalability of PDO generation. Integration of PDSOs with other models, such as organ-on-a-chip systems, can enhance their functional complexity and translational potential. Automation and robotics can facilitate high-throughput drug screening using large-scale PDO libraries, enabling rapid identification of potential therapeutic candidates for sarcomas. Continued research also enables the identification of new biomarkers and therapeutic targets specific to sarcomas, aiding in diagnosis, prognosis, and targeted therapy development. Furthermore, combining PDOs with technologies like single-cell sequencing and liquid biopsies can provide comprehensive molecular profiling and real-time monitoring of treatment response. The use of PDOs in conjunction with patient-specific genetic information holds great promise for personalized medicine in sarcoma management. By integrating genomic data and drug screening results from PDOs, tailored treatment strategies can be developed for individual patients, guaranteeing the quality of the usage of PDOs for research applications and increasing clinical translatability. Moreover, continued research has the potential to drive translational applications, paving the way for clinical trials and the validation of novel therapies. Personalizing treatment options based on an individual patient’s PDO characteristics can improve outcomes and reduce exposure to ineffective therapies. Standardized methods and validated clinical assays for functional testing in routine practice will be of utmost importance. More prospective data from clinical trials using PDOs to orient and guide clinical treatment are necessary. Addressing current limitations and challenges associated with sarcoma PDOs requires ongoing research and development. Although there are challenges and limitations to overcome, advancements in standardization, scalability, and integration with other platforms will further enhance the utility of sarcoma PDOs. Ultimately, the integration of PDOs into clinical practice has the potential to revolutionize sarcoma management and significantly improve patient outcomes. PDOs not only shed light on the biology of sarcoma but also improve the quality of life.

## Figures and Tables

**Figure 1 cancers-15-04339-f001:**
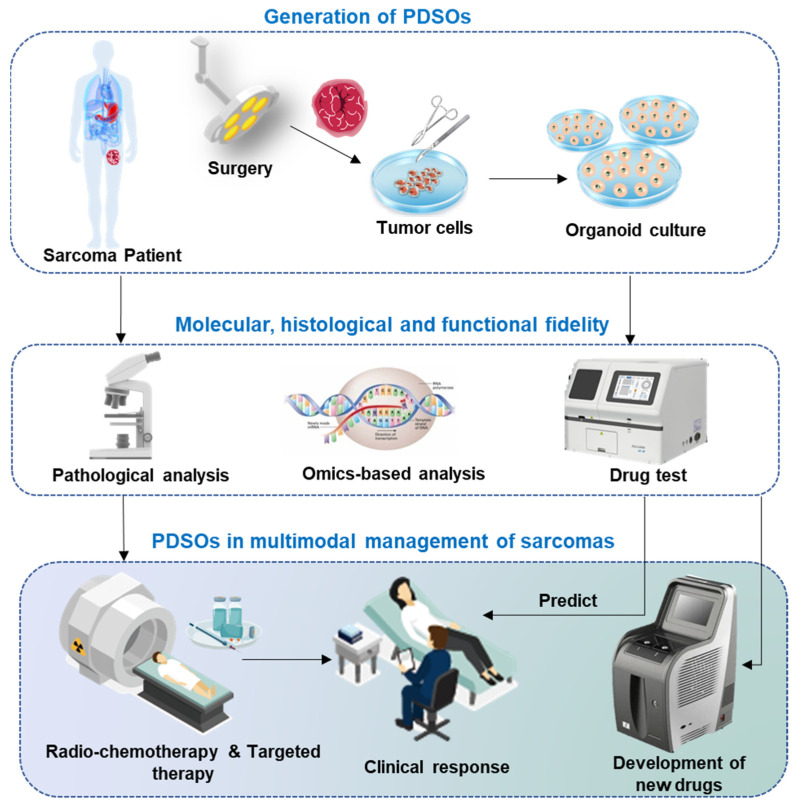
Schematic of the establishment of PDSOs and their applications. Sarcoma cells obtained from patients are grown into tumor organoids using scaffold-free, scaffold-based, and xenograft-based techniques. These patient-derived organoids realistically replicate the biological, histological, and functional characteristics of the primary tumor. By accurately recapitulating the complexity of the tumor microenvironment, PDSOs provide a promising approach to unravel sarcoma biology, discover novel therapeutic targets, and ultimately enhance patient outcomes.

**Figure 2 cancers-15-04339-f002:**
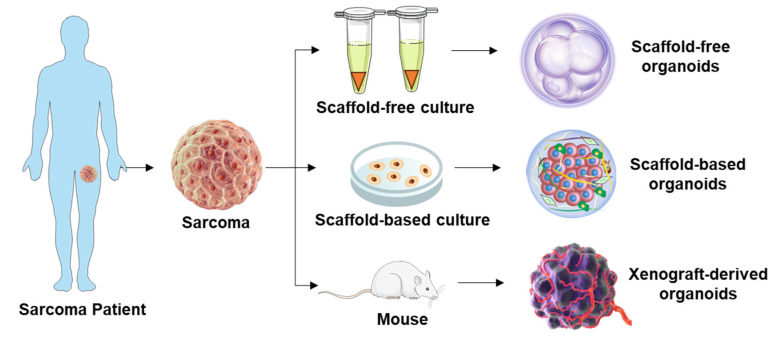
Generation of patient-derived organoids. The formation of organoids can be achieved through scaffold-free, scaffold-based, and xenograft-based methods.

**Figure 3 cancers-15-04339-f003:**
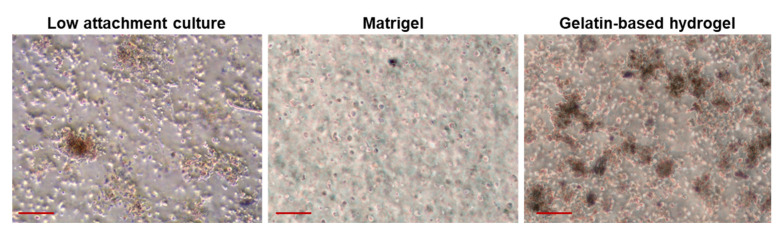
Organoids in different materials. The osteosarcoma organoids, derived from a 42-year-old female patient, were cultured in a low attachment culture dish, Matrigel, or 3% gelatin-based hydrogel (from left to right) for 7 days. Scale bar = 100 µm.

**Figure 6 cancers-15-04339-f006:**
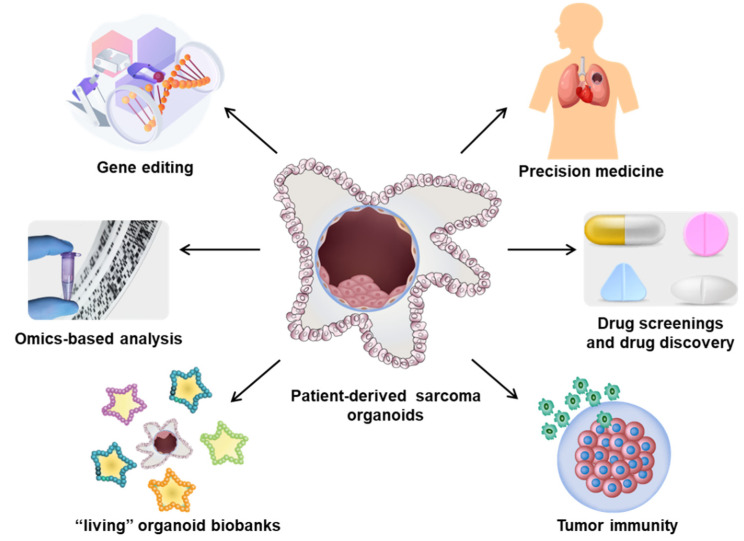
The application of PDSOs in personalized multimodal management. PDSOs have already been employed to establish dynamic and functional organoid biobanks, which can undergo various drug screenings for efficacy evaluations and drug discovery validations. Moreover, organoids have proven valuable in investigating both inter- and intra-tumor heterogeneity through the analysis of mutational signatures, gene expression patterns, and proteomics. Additionally, gene editing techniques can be effectively applied in PDSOs for disease modeling and fundamental research purposes. Furthermore, PDSOs provide an optimal platform for conducting research in the field of tumor immunology.

**Table 2 cancers-15-04339-t002:** Cell populations in sarcoma microenvironment.

Cell Category	Example of Cell Type	Contribution to Cellular Heterogeneity
Differentiated cells	Osteoblast Chondrocyte Smooth muscle cells	Exhibit varying degrees of differentiation.
Cancer stem cells		Have the ability to self-renew and differentiate into various cell types within the tumor. Play a role in tumor initiation, progression, metastasis, and treatment resistance.
Stromal cells	Fibroblasts Immune cells Endothelial cells	Secrete various growth factors, cytokines, and extracellular matrix components.

## Data Availability

The data presented in this study are available in this article.

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
