# Peer review of "Patient-Derived Organoids as a Promising Tool for Multimodal Management of Sarcomas"

_cancers, 2023, doi:10.3390/cancers15174339_

Round 1

Reviewer 1 Report

The authors herein present a very comprehensive review article describing the role of patient-derived sarcoma organoids in treating sarcomas. The various ways of establishing them, their specific advantages, advancements and limitations are all described in detail with examples from different literature sources. The overall content definitely gives abundant information on the role of them in improving the patient outcome and therefore the context is promising and clinically relevant. Authors have also used a simple way of describing the content which has shown improved clarity and readability.

There are some concerns which can be easily addressed by the authors and these are given below. Specifically, the lines showing the advantages of PDSOs appeared to be very redundant in almost every section. While it is acknowledged that authors intend to re-emphasize the benefits of PDSOs, redundant lines must be omitted and the article should highlight the key advantages in the relevant sections. Additionally, many examples describing studies have not been specified with details and therefore seems to be incomplete. Please find the comments below:

1.     The abbreviation of Patient-derived sarcoma organoids has been indicated as PDO in abstract and then PDSO in the later sections. Please correct this.

2.     Introduction, page 2, paragraph 2, line 78: “suggest the necessitate of…limitations” – is confusing and grammatically incorrect.

3.     Figure 1: it is quite difficult to connect the links between three figure panels here. Hence it is recommended to add more labels and captions in the figure to help explaining the connections and also provide brief description as figure legend.

4.     Table 1- explant culture: what is the proper usage?

5.     Figure 2- is this a figure taken from published content? If so, the authors need to specify that with copyright permission as well. There is not much information provided in lines 178-181 (page 5) too. What kind of hydrogel was used? What cells/ specific sarcoma? There needs to be extensive information to explain the figure and the important findings to define the figure data should be captured in the legend too. There should also be scalebar shown in both images.

6.     Page 6 , lines 196-198: Xu’s findings are also incomplete and there should be more specific information to support discussion. For example: include findings such as sarcoma type, type of PDSO used, details of animals etc. Please make sure that this is included throughout the manuscript for all discussed examples.

7.     Page 12, lines 249-258: these lines do not show any references.

8.     Pages 12-13, subsections on molecular, histological, and functional fidelity of PDSOs- lines 263-301: there needs to be supporting references throughout these sections. Surprisingly, there are no references to support all these statements!

9.     Page 14, line 365: indicate the clinical trial numbers.

10.  Page 15, line 400: ‘loss of this tumor suppressor in a’: should include ‘tumor suppressor gene’.

11.  Page 15, lines 417-418: ‘that was efficacious but toxic (ie, doxorubicin)’- very confusing statement. Authors need to redraft this line to convey the right scientific finding.

12.  I would also suggest categorizing the article to subsections by numbering properly.

13.  The authors must read through the article again and make sure that there is no redundancy of statements. As indicated in general comments, the advantages seem to be highlighted many times with the same words and statements throughout the article.

14. The current article only includes one schematic and one figure. Authors need to include more figures in the current version. This is a clinically relevant subject and several examples have been discussed in detail. Therefore, authors can easily include some reproduced images (at least 3-4) showing results from published examples discussed in the paper with adequate copyright permissions. Another suggestion, would be to be include schematics or pathways to describe some findings. 

Minor editing. Specific comments given above.

Author Response

We extend our sincere gratitude for your thoughtful and comprehensive review of our paper titled "Patient-derived Organoids as a Promising Tool for Multimodal Management of Sarcomas." Your valuable insights and constructive feedback have immensely contributed to enhancing the quality and clarity of our manuscript. We have carefully considered your comments and have incorporated the suggested revisions to address the concerns and strengthen the arguments (details have been listed below). We are confident that your insights will significantly enhance the quality and relevance of our paper, and we look forward to sharing the improved version with you.

Thank you for your time, dedication, and support.

Specific Comments:

  1. The abbreviation of patient-derived sarcoma organoids has been indicated as PDO in abstract and then PDSO in the later sections. Please correct this.

Response:

Thanks! We have corrected them in the revised manuscript.

Specific Comments:

  1. Introduction, page 2, paragraph 2, line 78: “suggest the necessitate of…limitations” – is confusing and grammatically incorrect.

Response:

We apologize for grammatically incorrect. We have rephrased it to “The heterogeneity of sarcomas within individual patients and the development of treatment resistance over time underscore the necessity for implementing multimodal management strategies to address the therapeutic limitations”. Please see Page 2, lines 77-79.

Specific Comments:

  1. Figure 1: it is quite difficult to connect the links between three figure panels here. Hence it is recommended to add more labels and captions in the figure to help explaining the connections and also provide brief description as figure legend.

Response:

Thanks! We have incorporated additional labels and captions into Figure 1 in the revised manuscript, along with a concise description provided in the figure legend.

Specific Comments:

  1. Table 1- explant culture: what is the proper usage?

Response:

We have removed Table 1 as it appears to be redundant.

Specific Comments:

  1. Figure 2- is this a figure taken from published content? If so, the authors need to specify that with copyright permission as well. There is not much information provided in lines 178-181 (page 5) too. What kind of hydrogel was used? What cells/ specific sarcoma? There needs to be extensive information to explain the figure and the important findings to define the figure data should be captured in the legend too. There should also be scalebar shown in both images.

Response:

We have replaced these images with alternative ones of superior quality (revised Figure 3) and provided extensive information in the legend.  In addition, we have added scale bars to these images (revised Figure 3). 

Specific Comments:

  1. Page 6, lines 196-198: Xu’s findings are also incomplete and there should be more specific information to support discussion. For example: include findings such as sarcoma type, type of PDSO used, details of animals etc. Please make sure that this is included throughout the manuscript for all discussed examples.

Response:

We have provided more detailed information on the revised manuscript as suggested, as shown below:

Xu et al. studied xenograft PDO using thirty-six bone sarcoma and six tissue sarcoma (details listed in Table 1), and they showed an overall higher engraftment rate of 73.8% compared to most other studies, but they used a triple immunodeficient NCG mice. Please see Page 6, lines 201-204.

Specific Comments:

  1. Page 12, lines 249-258: these lines do not show any references.

Response:

Thanks. We have added the according references on the revised manuscript. Please see Page 13, lines 254-261.

Specific Comments:

  1. Pages 12-13, subsections on molecular, histological, and functional fidelity of PDSOs- lines 263-301: there needs to be supporting references throughout these sections. Surprisingly, there are no references to support all these statements!

Response:

Thanks. We have added the according references on the revised manuscript. Please see Page 13, lines 267-291.

Specific Comments:

  1. Page 14, line 365: indicate the clinical trial numbers.

Response:

We have included the clinical trial numbers, as shown below:

“……drug synergy has been shown in 40 patients with multiple myeloma in clinical trials (NCT 01246063).” Please see Page 16, lines 381-382.

Specific Comments:

  1. Page 15, line 400: ‘loss of this tumor suppressor in a’: should include ‘tumor suppressor gene’.

Response:

We have added the tumor suppressor gene, as shown below:

A faster growth rate and invasiveness could result from the loss of this tumor suppressor CDKN2A in a patient-derived myxofibrosarcoma. Please see Page 17, line 417.

Specific Comments:

  1. Page 15, lines 417-418: ‘that was efficacious but toxic (ie, doxorubicin)’- very confusing statement. Authors need to redraft this line to convey the right scientific finding.

Response:

We have redrafted the sentences to clarify the finding, as shown below:

For example, in one case, a 16-year-old male diagnosed with synovial sarcoma initially received a post-surgical systemic adjuvant therapy with doxorubicin37. The PDSOs model allowed clinical improvement by suggesting liposomal-doxorubicin as a better agent that have same efficacious but lower toxicity as compared to doxorubicin37. The researchers applied sequentially to the patient, resulting in over 20 months of event-free survival37.

Specific Comments:

  1. I would also suggest categorizing the article to subsections by numbering properly.

Response:

We have categorized the article with numbers accordingly as suggested. All changes have been highlighted in the article.

Specific Comments:

  1. The authors must read through the article again and make sure that there is no redundancy of statements. As indicated in general comments, the advantages seem to be highlighted many times with the same words and statements throughout the article.

Response:

We have removed all redundancy in the revised manuscript.

Specific Comments:

  1. The current article only includes one schematic and one figure. Authors need to include more figures in the current version. This is a clinically relevant subject and several examples have been discussed in detail. Therefore, authors can easily include some reproduced images (at least 3-4) showing results from published examples discussed in the paper with adequate copyright permissions. Another suggestion, would be to be include schematics or pathways to describe some findings.

Response:

            We thank the reviewer for the expert comments and suggestions.  We have added another 4 diagrams of the proposed examples in the revised manuscript as suggested (revised Fig. 2,4,5,6).

Reviewer 2 Report

Thank you for your valuable manuscript about human sarcoma organoid models. I thought the manuscript was well written about present and future of sarcoma organoids. Basically, I thought the review study is acceptable for the journal.

And I appreciate your citation of the study about human epithelioid sarcoma organoids. We also recently published about human organoid of malignant giant cell tumor, thus I am glad to check about it. (https://doi.org/10.1016/j.jbo.2023.100486)

There are a few points to revise as followings.

In line 57, authors used “malignant fibrous histiocytoma”. Current version of WHO calssification, “undifferentiated pleomorphic sarcoma” is used. Probably, cited study used the old name. It is better to add annotation about it.

In line 162, “we will discuss”, please change it to “we discussed”.

Author Response

We thank the reviewers for the expert comments and suggestions. Additionally, we extend our appreciation to the reviewer for providing us with the latest research paper that aligns with our focus. We had added it in our paper, please see reference 50 and Table 1.

Specific Comments:
There are a few points to revise as followings.

In line 57, authors used “malignant fibrous histiocytoma”. Current version of WHO calssification, “undifferentiated pleomorphic sarcoma” is used. Probably, cited study used the old name. It is better to add annotation about it.

Response:

We have annotated about it, as showed in line 56- 58, “while leiomyosarcoma and malignant fibrous histiocytoma (currently classified as undifferentiated pleomorphic sarcoma by WHO) are more frequent in adults.” Please see Page 2, lines 57-58.

Specific Comments:
In line 162, “we will discuss”, please change it to “we discussed”.

Response:

We have changed “we will discuss” to “we discussed”. Thank you for pointing out grammar error. please see Page 5 in line 165.

Round 2

Reviewer 1 Report

Authors have now revised the manuscript thoroughly based on the suggestions raised in the previous review. The overall quality of the revised manuscript has substantially increased. Authors have answered all the questions and appropriate corrections have been made. Additional figures have also been added, and sufficient literature support has also been provided as suggested. There are no outstanding concerns at this point and this revised version can be considered for acceptance.